# Item-level analyses reveal genetic heterogeneity in neuroticism

Mats Nagel[1], Kyoko Watanabe[2], Sven Stringer [2], Danielle Posthuma [1,2] & Sophie van der Sluis[1]

Genome-wide association studies (GWAS) of psychological traits are generally conducted on (dichotomized) sums of items or symptoms (e.g., case-control status), and not on the individual items or symptoms themselves. We conduct large-scale GWAS on 12 neuroticism items and observe notable and replicable variation in genetic signal between items. Within samples, genetic correlations among the items range between 0.38 and 0.91 (mean $r_g = .63$), indicating genetic heterogeneity in the full item set. Meta-analyzing the two samples, we identify 255 genome-wide significant independent genomic regions, of which 138 are item-specific. Genetic analyses and genetic correlations with 33 external traits support genetic differences between the items. Hierarchical clustering analysis identifies two genetically homogeneous item clusters denoted depressed affect and worry. We conclude that the items used to measure neuroticism are genetically heterogeneous, and that biological understanding can be gained by studying them in genetically more homogeneous clusters.

[1] Department of Clinical Genetics, Section Complex Trait Genetics, Center for Neurogenomics and Cognitive Research, Amsterdam Neuroscience, VU Medical Centre, Amsterdam 1081 HV, The Netherlands. [2] Department of Complex Trait Genetics, Center for Neurogenomics and Cognitive Research, Amsterdam Neuroscience, VU University Amsterdam, Amsterdam 1081 HV, The Netherlands. Danielle Posthuma and Sophie van der Sluis contributed equally to this work. Correspondence and requests for materials should be addressed to D.P. (email: d.posthuma@vu.nl) or to S.S. (email: s.vander.sluis@vu.nl)

G WAS on psychological traits like major depressive disorder ($h^2_{SNP} = 0.06$)[1], intelligence ($h^2_{SNP} = 0.20$)[2] and neuroticism ($h^2_{SNP}$ range: 0.09–0.15)[3,4] have uncovered numerous associated variants. These genome-wide significant single-nucleotide polymorphisms (SNPs), however, explain only a small portion of the twin heritability ($h^2_{twin} = 0.40$, 0.54, and 0.47, respectively)[5]. The small individual SNP effects support the infinitesimal model, which assumes the involvement of many genetic variants (i.e., polygenicity), each of (very) small effect, such that large sample sizes are required to robustly detect them.

Genetic studies on psychological traits often adopt phenotypic composite scores, such as a sum-score or binary case–control status, which summarize information contained in multiple items or symptoms. In psychological research, composite scores have proven useful, e.g., in directing therapeutic intervention, and in predicting future school/job performance. However, the items or symptoms collectively operationalizing one trait can be very diverse in nature. For instance, in personality inventories for neuroticism, items vary from 'feeling miserable' and 'experiencing mood swings' to 'feeling guilty' and 'worry too long after an embarrassing experience'. Similarly, diagnostic symptoms for major depressive disorder (MDD) vary from 'increase in appetite', 'irritable mood', and 'fatigue' to 'insomnia', 'excessive guilt', and 'suicidal ideation'[6]. Previous studies[7–9] have shown that items or symptoms underlying the same composite score can indeed differ considerably with respect to e.g. their relations to external risk factors, their impact on impairment, and their underlying biology. In the context of gene-finding studies, power to detect associated variants is potentially lost when the summed items or symptoms are biologically heterogeneous. Specifically, the use of composite scores in gene-finding studies directs the focus of analysis to those variants that affect the majority of aggregated items, i.e., 'global variants'. The genetic signal of "local" variants, affecting only one or a few of the aggregated items, is severely diluted[10]. That is, if summed items or symptoms are genetically heterogeneous, then GWAS analysis of their sum may yield a mix of diluted signals, which will bear resemblance to the infinitesimal model.

One can investigate the genetic homogeneity of items or symptoms underlying a sum-score or a case–control status by studying the genetics of the individual items. If items prove to be genetically heterogeneous, identification of more homogeneous subsets or clusters of items may be expedient to optimize the statistical power to disentangle genetic determinants. Indeed, detection of local variants, affecting, e.g., only one cluster of items but not others, might facilitate biological understanding.

Here, we use data on 12 dichotomous neuroticism items (Supplementary Table 1; Eysenck Personality Questionnaire—Revised Short Form[11]), to investigate whether the items used to operationalize neuroticism are genetically homogeneous. Given its high correlation with depression ($r = 0.58$; genetic correlation $r_g = 0.60$) and anxiety ($r = 0.75$; $r_g = 0.77$)[12,13], neuroticism is considered an important phenotype in psychiatric genetic research[14]. Indeed, neuroticism items often resemble items used to measure (anxious) depression (see Supplementary Table 1 for a substantive overview). For the present study, data was obtained from two samples of the UK Biobank cohort[15] (sample 1: first UK Biobank release, available since 2015; sample 2: data added in the second UK Biobank release, available since July 2017; item-specific $N$ range sample 1: 106,218–109,017; sample 2: 260,083–266,896; see Methods; Supplementary Tables 1, 2). We use sample 2 to replicate basic findings of sample 1, and then meta-analyze the results of both samples to compare item-specific genetic signals to the signal obtained in genetic analysis of the sum-score. Our study demonstrates that item-level analyses supplement genetic sum-score analysis, as the 12 neuroticism items often show only moderate genetic overlap. Furthermore, we identified two genetically distinct item clusters, which may prove useful targets of investigation in future genetic analyses.

## Results

**Phenotypic analyses**. Phenotypic analyses of sample 1 showed that the 12 neuroticism items correlated positively with each other (.17–.54, Supplementary Fig. 1), and with the weighted sum-score ('sum-score' henceforward; .51–.68, see Methods for information on how the sum-score was constructed). Associations of all 12 items and the sum-score with external (demographic/psychological) variables had largely the same sign, but occasionally showed considerable differences in magnitude (Supplementary Tables 3, 4; Supplementary Figs 2, 3).

**Genetic correlations between items**. We performed 13 GWASs (12 items + sum-score) on sample 1 and applied bivariate LD score regression[16,17] on the summary statistics to compute genetic correlations ($r_g$) between all 13 phenotypes (upper triangle Fig. 1; Supplementary Data 1; see Methods for QC and technical details). Between items, $r_g$'s ranged from a low 0.38 (IRR/WORR-EMB; s.e. = 0.048) to a high 0.91 (MOOD/FED-UP; s.e. = 0.030, mean $r_g = .64$), and none of the 95% confidence intervals included 1. These results indicate genetic heterogeneity in the full item set. To replicate these findings, we conducted the same 13 GWASs in sample 2 (lower triangle Fig. 1). The $r_g$'s between the exact same items measured in both samples were all close to 1 (diagonal Fig. 1; all 95% confidence intervals included 1) indicating that the genetic signal is highly similar across the two samples. In addition, the correlation between inter-item $r_g$'s in samples 1 and 2 was 0.97, confirming a highly similar $r_g$ pattern across the two samples.

**GWAS meta-analyses**. Having replicated the genetic heterogeneity in the set of neuroticism items, we moved on, for reasons of robustness, to conduct meta-analysis on the GWAS results of the two samples for all 12 items ($N$ range: 366,301–375,913) and the sum-score ($N = 380,060$) to allow comparison between item-level GWAS results and results obtained in GWAS of the sum-score (Supplementary Table 5).

In the 13 phenotypes, a total of 16,825 SNPs were genome-wide significant (GWS: $P < 5 \times 10^{-8}$), of which 2,474 were located in inversions on chromosomes 8 and 17, as reported previously for the neuroticism sum-score[3,18,19]. All GWS variants had the same direction of effect in both meta-analyzed samples. The 16,825 detected variants tagged 493 lead SNPs (see Methods for definition of lead SNPs), mapping to 255 independent genomic regions (based on clumping using an $r^2$ threshold of 0.1; Supplementary Datas 2–17; Supplementary Figs. 4–17; see Methods; http://fuma.ctglab.nl/20). Of the total 255 regions, 117 were GWS for the sum-score, and 6 to 44 regions (median = 32) were GWS for individual items. Genetic signal varied considerably between items, with some (e.g., LONE, SUF-NERV) showing only a few GWS associations, while others (e.g., IRR, MIS, MOOD, NERV-FEEL, WORRY) showed > 35 GWS genetic regions. Furthermore, of all 255 regions, 138 were GWS in item-level analyses only and not for the sum-score (Supplementary Data 2), and 42 were GWS in the sum-score analysis while no GWS association was observed for any of the items.

The 493 lead SNPs implicated 908 genes through positional mapping, eQTL mapping and/or chromatin interactions (see Methods; Supplementary Data 18–30). Of these 908 genes, 473 were only associated to one or more individual item(s) and not to the sum-score[3,4,21,22] (Supplementary Fig. 18a).

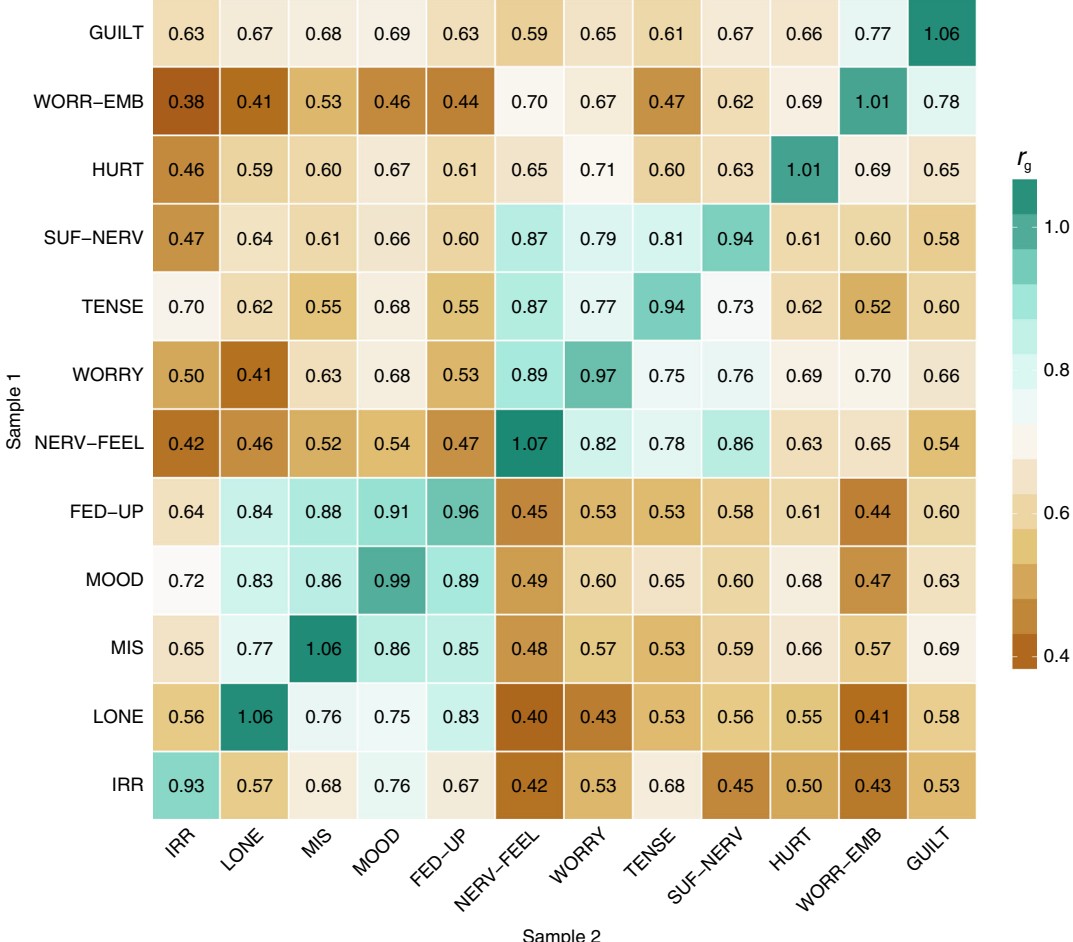

**Fig. 1** Genetic correlations between the 12 neuroticism items. Genetic correlations between the 12 neuroticism items within both samples (sample 1 is presented in the upper triangle and sample 2 in the bottom triangle). The genetic correlations of the same items measured in both samples are presented on the diagonal. All reported genetic correlations, computed with bivariate LD score regression, are significantly different from zero after Bonferroni correction ($P < 2.22 \times 10^{-4}$). For the genetic correlations between the same items (diagonal) all 95% confidence intervals included 1, emphasizing concordance in genetic signal across samples. For all other, cross-phenotype genetic correlations, none of the 95% confidence intervals included 1, indicating that all genetic correlations deviated from unity (Supplementary Data 2). See Supplementary Table 1 for a description of the item labels

Liability-scale SNP-based heritability estimates ($h^2_{SNP}$) for the 13 phenotypes, established using LD score regression (LDSC)[17], ranged from 8% (s.e. = 0.0042) to 12% (s.e. = 0.0054) (Supplementary Fig. 19), and all estimates deviated significantly from 0.

In studying overlap in genetic signal between the 13 phenotypes (Methods; Supplementary Note 1), sign concordance tests showed that effect signs of top SNPs were largely concordant across the 13 phenotypes (Supplementary Fig. 20; Supplementary Data 31). However, Fisher's exact tests revealed that the exact top SNPs overlapped only moderately between the 13 phenotypes (Supplementary Fig. 21; Supplementary Data 32), indicating that the most strongly associated SNPs differed considerably between individual measures.

**Gene-based analysis**. By combining the genetic signal of all SNPs in a gene (while accounting for LD between SNPs), gene-based analyses can implicate genes that may go unnoticed in SNP-based analyses. Using the $P$-values from the GWA meta-analyses (see Methods), we conducted gene-based analyses in MAGMA[23] on all 13 phenotypes. In total, 654 genes reached GWS ($P < 2.73 \times 10^{-6}$), of which 388 overlapped with genes implicated in SNP-based analyses (Supplementary Datas 33–46; Supplementary Figs. 18b, 22).

275 of the MAGMA genes were associated to the neuroticism sumscore, whereas 29 (LONE) to 172 (WORRY) genes (median = 96) were GWS for individual items. Of the total of 654 genes, 379 were item-specific, i.e., only reached GWS in one or more individual items, indicating genetic heterogeneity between the 12 items, and between the items and their sum (Supplementary Data 33; Supplementary Fig. 18c).

Overall, the results from the gene-based analyses suggest a role for both 'global' genes (i.e., affecting (almost) all items and the composite score) and 'local' genes, (i.e., affecting only one or a subset of items). In Table 1, a selection of such global and local genes is highlighted. Combined, SNP-based and gene-based analyses on all 13 phenotypes implicated 1,247 genes, of which 651 were not identified in analysis of the sum-score (Supplementary Figs. 18d).

**Gene-set analysis**. While associations to individual SNPs or genes may differ between specific items and their summed score, it is possible that similar gene-sets and pathways are implicated. To test this, we used the results of the gene-based analyses as input for MAGMA's gene-set analysis[23]. For each of the 13 phenotypes we tested 7,244 gene-sets, comprising curated and gene ontology

**Table 1 Selection of interesting genes identified in gene-based analyses of the 13 neuroticism phenotypes**

| Gene | CHR | Position | Implicated by | Notes |
|---|---|---|---|---|
| SPPL2C | 17 | 43,922,256 | All 13 phenotypes | Only gene GWS associated to all 13 neuroticism phenotypes. Located in known inversion on chromosome 17. Associated to e.g. red blood cell count and Parkinson's disease[47,48]. |
| MAPT | 17 | 43,971,702 | 11 of 13 phenotypes | Promotes microtubule assembly and stability. Previously linked to intelligence[49] and neurodegenerative disorders (e.g. Parkinson's and Alzheimer's disease[50,51]). |
| DRD2 | 11 | 113,280,317 | 11 of 13 phenotypes | Previously associated to neuroticism sum-score[3]. D2-receptors are thought to be involved in reward processing. |
| GRM8 | 7 | 126,078,652 | 8 items + sum-score | Not associated to neuroticism before. Has been associated to the response to selective serotonin reuptake inhibitors (SSRI) in depressed individuals[52]. |
| SORCS3 | 10 | 106,400,859 | Items in the depressed affect cluster and sum-score, but not items in the worry cluster | Signal is driven primarily by 3 items (MIS, MOOD, FED-UP), all part of the depressed affect cluster. SORCS3 has been associated to depression[3], and is involved in neuropeptide receptor activity. |
| CADM2 | 3 | 85,008,133 | Items in the worry cluster and sum-score, but not items in the depressed affect cluster | Most strongly associated gene for the worry cluster ($P = 2.24 \times 10^{-23}$), and GWS for all items in this cluster. Linked to BMI in multiple studies[53,54] (see genetic correlations between both clusters and BMI in Fig. 3). |
| Region A | 3 | 49.215–50.226 Mb | Depressed affect items | Genes in this region (e.g., C3orf84, RHOA, MST1, APEH) were previously linked to e.g. Crohn's disease[55] and blood protein levels[56]. A subset of the genes in this region (e.g. IP6K1, CAMKV, SEMA3F) have been associated with educational attainment[57]. |
| Region B | 3 | 50.264–53.080 Mb | Worry items | Genes in this region (e.g. ALAS1, STAB1, GNL3, ITIH4, TMEM110) have been linked to schizophrenia and autism spectrum disorder in earlier studies[58,59]. |

All genes that were GWS in gene-based analyses in MAGMA[23] on all 13 neuroticism phenotypes are reported in Supplementary Data 33.
CHR: Chromosome on which the gene is located.
Position: Start position of the gene in base pairs (for the regions A and B a start and end position is reported).
Depressed affect and worry refer to two separate sets of 4 items each, i.e., LONE/MIS/MOOD/FED-UP, and NERV-FEEL/WORRY/TENSE/SUF-NERV, respectively.
See Supplementary Table 1 for a description of the item labels.

(GO) gene-sets derived from MsigDB[24], and 53 tissue expression profiles[25]. The sum-score was significantly ($P < 0.05/7,297 = 6.85 \times 10^{-6}$) associated to 9 GO gene-sets, whereas 8 GO gene sets were associated to one or more individual items, but not to the sum-score (Supplementary Data 47; Fig. 2). Noteworthy, 4 out of 9 GO gene-sets that were found to be associated to the sum-score were also associated to individual items, indicating that the sum-score and the individual items at least in part implicate the same biological processes.

The neuroticism sum-score was associated with genes expressed in 6 brain tissue types, all of which were also associated to at least one of the individual items. In addition, 2 brain tissue types, amygdala and caudate basal ganglia, were only identified in item-level gene-set analyses. Overall, while some gene-sets were implicated by multiple sources (Fig. 2), the gene-set analyses also confirmed the presence of genetic heterogeneity between the 12 neuroticism items and the sum-score.

**Functional annotation**. A comparison of functional consequences, chromatin state, and regulatory functions of all the SNPs in LD with one of the independent significant SNPs identified in item-level versus sum-score analysis is provided in Supplementary Tables 6–8 and Supplementary Fig. 23.

For the sum-score we identified 36 GWS exonic non-synonymous SNPs (ExNS), of which 29 overlapped with the 134 ExNS SNPs identified in item-level analyses (Supplementary Data 48). 10 of these overlapping ExNS SNPs are located in a well-known inversion on chromosome 17[3]. Overall, 105 ExNS SNPs were specifically associated with one or more of the individual items and went unnoticed in sum-score analysis, showing that item-level analyses superadded to the identification of SNPs that are highly likely to have functional consequences. As

an example, we highlight two ExNS SNPs, not GWS for the neuroticism sum-score[19], that may be viable candidates for functional follow-up. Of all ExNS SNPs associated to neuroticism items, rs45510500, located in exon 42 of KIAA1109, had the highest CADD score (35). rs45510500 is a missense mutation that leads to an amino acid change of Arginine to Tryptophan. The second SNP, rs3130618 in exon 3 of GPANK1 with a CADD score of 34, is a missense mutation resulting in an Arginine to Leucine change. rs3130618 has a regulome database score of 1f, implying that it is likely to affect binding and to affect expression of a gene target.

**Genetically homogeneous item clusters**. We performed a hierarchical clustering analysis on the inter-item $r_g$'s of sample 1 to see whether the items could be grouped into genetically homogeneous clusters (Fig. 1). This analysis revealed two clusters (1: LONE/MIS/MOOD/FED-UP 2: NERV-FEEL/WORRY/TENSE/SUF-NERV), with mean $r_g = .84$ (range: 0.77–0.91) within clusters, and a much lower mean $r_g = .59$ (range: 0.38–0.77) between items from different clusters. The identified clusters coincide with the dimensions depressed affect and worry previously identified in factor analysis of the full EPQ-R neuroticism scale[26].

We then applied the item order that emerged from the cluster analysis on sample 1 to the genetic correlational results from sample 2, and observed the same two item clusters (lower triangle of Fig. 1), with very similar within and between item $r_g$'s (overall mean $r_g = 0.62$; range $r_g$'s: 0.40–0.89; within clusters mean $r_g = 0.80$, range: 0.73–0.89; outside of clusters mean $r_g = 0.58$, range: 0.40–0.78). The genetically homogenous clusters thus replicated robustly within the UKB sample. All items had a relatively high $r_g$ with the sum-score (range: 0.73–0.89), but items assigned to clusters correlated even higher with their respective cluster, while

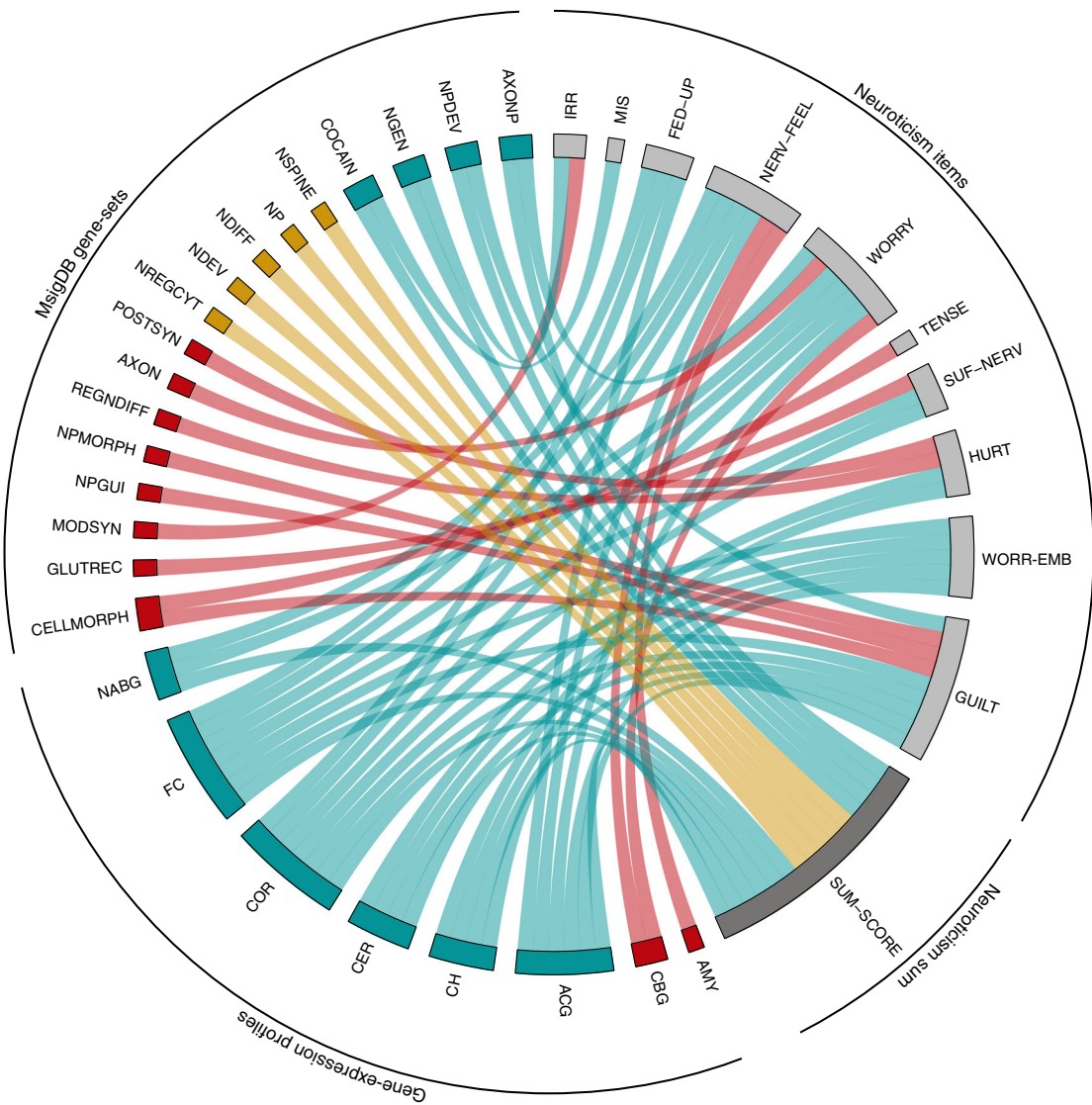

**Fig. 2** Circle plot of statistically associated gene-sets of the 7,297 gene-sets tested (MsigDB gene-sets and gene-expression profiles). This circle plot presents all gene-sets that were significantly associated (Bonferroni corrected $P$-value threshold: $0.05/7,297 = 6.85 \times 10^{-6}$) to at least one of the individual neuroticism items or to the sum-score. Each tract connects a gene-set to the phenotype with which it is associated. Tracts colored in red indicate gene-sets that were associated to individual items, but not to the sum-score. In contrast, tracts colored in yellow indicate gene-sets that were associated to the neuroticism sum-score, but not to any of the individual items. Blue tracts refer to gene-sets that were found to be associated to the sum-score and at least one of the individual items. See Supplementary Data 47 for a description of the gene-sets included in this figure. See Supplementary Table 1 for a description of the item labels

$r_g$'s with the opposite cluster were considerably lower (difference > .20), confirming genetic heterogeneity within the full item set (Supplementary Fig. 24; Supplementary Data 1). Genetic analysis of such genetically homogenous clusters may reveal genetic signals that are important for multiple items in one cluster but not for items in the second cluster, as we recently showed in ref. 19.

**Genetic correlations with external traits**. Neuroticism is linked to numerous social, behavioral, and psychiatric traits[4]. We therefore calculated $r_g$'s of the 12 items, the sum-score, and the two identified item clusters with 33 external traits (Supplementary Data 49) using cross-trait LD score regression[16].

In line with previous findings for neuroticism sum-scores[3,4,21], all items correlated negatively with subjective well-being (−.67 to −.45, sum-score: −.65), and positively with major depressive disorder (.35−.66, sum-score: .65), depressive symptoms (.42−.83, sum-score: .79), and anxiety disorders (.51−.78, sum-score: .76; Fig. 3; Supplementary Data 50). These strong positive $r_g$'s are presumably at least partly due to the substantial overlap in content between neuroticism items on the one hand and depression/anxiety items on the other (see Supplementary Table 1), which reduces the operational distinctness of these phenotypes.

In contrast, $r_g$'s of individual items to other external traits were not all in the same direction (e.g., IQ, BMI). Here, relying merely on sum-score information would be misleading: sum-score $r_g$'s

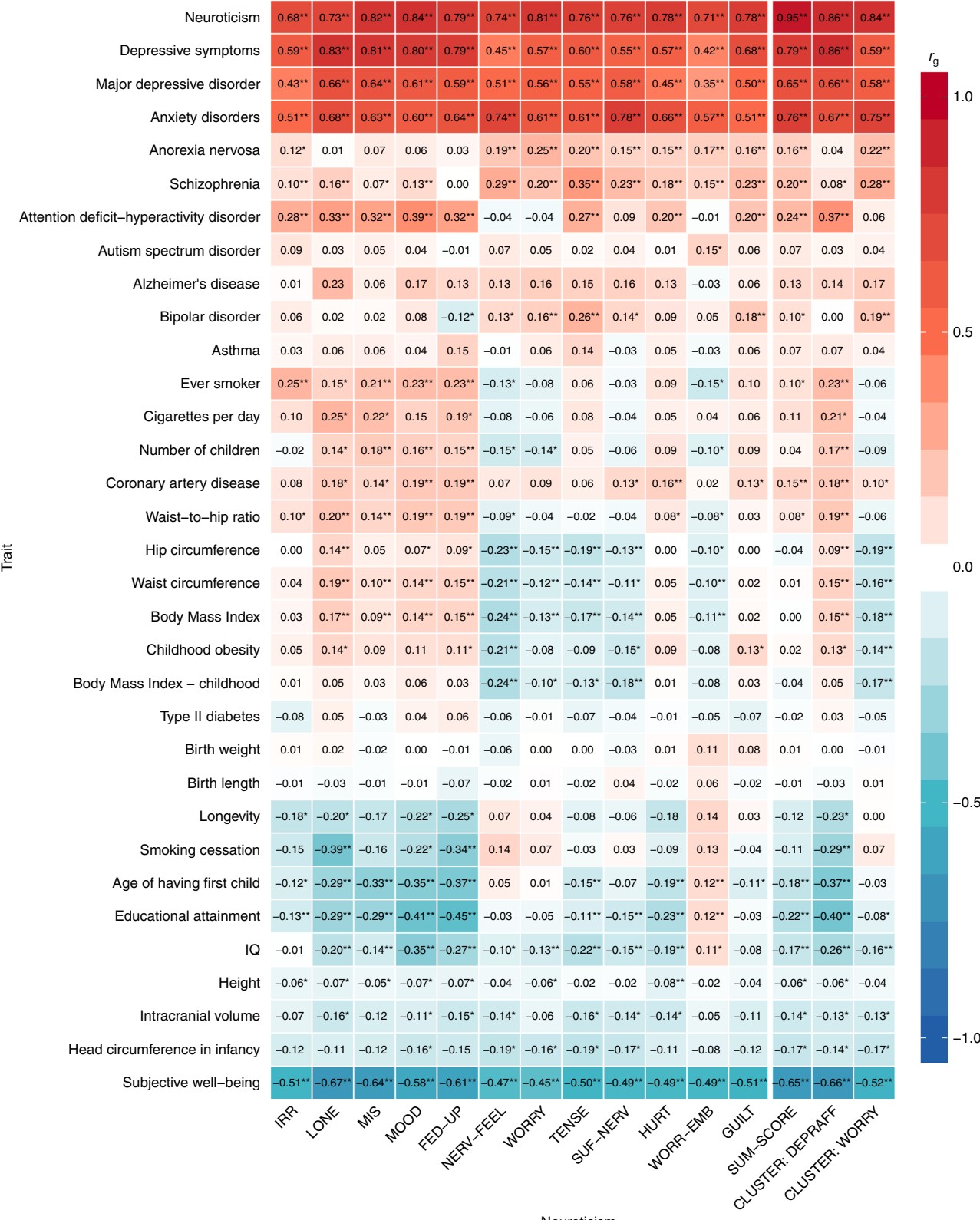

**Fig. 3** Genetic correlations of 12 neuroticism items, the sum-score and the two item clusters with 33 external traits. Genetic correlations of neuroticism items (see Supplementary Table 1 for a description of the item labels), sum-score, and the item clusters depressed affect (CLUSTER: DEPRAFF) and worry (CLUSTER: WORRY; x axis) with 33 other traits (y axis; Supplementary Data 49) computed with cross-trait LD score regression. P values and 95% confidence intervals for the $r_g$'s are provided in Supplementary Data 50. Exact values of the genetic correlations are shown in the cells. *$P < 0.01$; **Bonferroni corrected P-value threshold: $P < 1.01 \times 10^{-4}$ (0.05/(15 × 33))

can be small while item-specific $r_g$'s are substantial yet opposite ($r_g > |.17|$; see e.g., BMI, hip and waist circumference).

The magnitude of $r_g$'s associated with the items assigned to the clusters depressed affect and worry varied substantially for many traits (e.g., depressive symptoms, subjective well-being). Some of the traits showed genetic overlap with items of one of the clusters, but not with items from the other cluster (e.g., anorexia, schizophrenia, educational attainment, ADHD), whereas a few traits even showed GWS genetic correlations with items in both clusters yet in opposite directions (e.g., BMI, waist and hip circumference). Smoking related phenotypes (i.e., ever smoker, smoking cessation) showed significant correlations only to items of the depressed affect cluster, and not to items of the worry cluster. These results clearly indicate that biological insight may be gained by analyzing these genetically homogeneous clusters separately.

## Discussion

In summary, we have shown that items used to measure the personality trait neuroticism are genetically heterogeneous, with genetic overlap often being only moderate (i.e., $r_g < .60$), and we replicated this in a second sample. We then proceeded to conduct GWAS meta-analyses on all 12 neuroticism items and their sum-score in ≥366,276 individuals. We identified 138 item-specific loci implicating 1,247 genes, revealing considerable genetic variation between items. We identified two genetically distinct item clusters denoted depressed affect and worry. Within clusters, items were genetically strongly correlated, and items from different clusters showed distinct genetic correlational patterns to external traits.

The current findings motivate investigations into the genetic heterogeneity of items in other instruments used to gauge complex psychological traits. Neuroticism is only one of many psychological traits for which composite scores are calculated that are based on the aggregation of item or symptom scores. As a clinical example, consider the DSM-V diagnosis for major depressive disorder (MDD). This diagnosis is based on a list of 9 diverse symptoms (at least four of which are "aggregated" symptoms, reflecting problems on either end of the spectrum, e.g., 'insomnia or hypersomnia', 'increase or decrease in appetite', see Supplementary Note 2 for the full list of symptoms). To qualify for a depression diagnosis, at least 5 of these symptoms should be endorsed for at least 2 weeks, a procedure that can result in people with very different symptom profiles obtaining the same diagnosis[8,27]. In subsequently using the diagnostic status as dependent variable in GWAS, the assumption is that these symptoms are genetically similar. The phenotypic heterogeneity of the symptoms does, however, like with neuroticism, raise questions about their alleged genetic homogeneity. As yet, genetic heterogeneity between depression symptoms has only been addressed in the context of twin studies[28].

Similarly, many psychological instruments measuring quantitative traits are known to be phenotypically multidimensional. For instance, the multidimensionality of cognitive ability is evident in the many subscales characterizing renowned IQ tests like the Wechsler Adult Intelligence Scale (WAIS[29,30]). Twin and family studies show that these multidimensional phenotypes are also genetically multidimensional, and that dimension-specific genetic effects can be substantial[31–35].

In sum, while all these psychological instruments have proven useful in therapeutic settings and in, e.g., predicting school/job performance, the known phenotypic heterogeneity and multidimensionality begs the question whether an overall sum-score operationalization is expedient in gene finding studies, as there is generally no a priori reason to assume that the subscales, items, or symptoms are genetically similar. By studying the genetic architecture of individual subscales or items or symptoms, we can determine whether the apparent genetic complexity of psychiatric traits is (at least partly) introduced by our choice of operationalization. Item-level or symptom-level analyses need not become the standard, but they can inform investigation and construction of genetically homogeneous subsets, as we have shown in the present study.

Our study highlights the purpose and relevance of item or symptom-level analyses as a means to inspect genetic homogeneity, and to identify subsets of genetically similar items or symptoms that can confidently be summed and used as input in future gene-finding enterprises.

## Methods

**UK Biobank sample.** The UK Biobank Study is a major data resource, containing phenotypic measures from 503,325 participants and genetic data from 489,212 participants (July 2017 release)[15]. The data was released in two phases (May 2015 and July 2017), and based on this we created two separate samples on which we performed the GWA analyses.

Sample 1 consists of individuals for whom the data was released in May 2015 ($N = 110,328$; Supplementary Table 1), whereas sample 2 consists of all individuals that were added in the 2nd release (July 2017; $N = 270,178$; Supplementary Table 2). Written informed consent was obtained from all participants and the UK Biobank received ethical approval from the National Research Ethics Service Committee North West–Haydock (ref. 11/NW/0382). The current study was conducted under UK Biobank application number 16406.

For the analyses in this study, we excluded all individuals of non-Caucasian ancestry (based on genetic principal components). Therefore, principal components from the 1000 Genomes reference populations[36] were projected onto the called genotypes available in UK Biobank. Those participants whose projected principal component score was closest to the average score of the European 1000 Genomes (based on the Mahalanobis distance) were identified as European. We excluded European subjects with a Mahalanobis distance > 6 s.d. Additionally, subjects were filtered out based on relatedness, discordant sex, sex aneuploidy, and withdrawn consent. Finally, individuals were excluded from analyses if responses to more than 3 (of 12) of the neuroticism items were missing. The effective sample in our meta-analysis consisted of 380,506 individuals (205,556 females and 174,950 males; see Supplementary Table 5 for item-specific sample sizes). At the time of test completion, participants' age ranged between 40 and 73 ($M = 56.91$; s.d. = 7.93).

For both samples specified above, we used newly imputed data from the 2nd release, on ~96 million genetic variants. Imputation was performed by the UK Biobank, using a reference panel that included the UK10K haplotype panel as well as the Haplotype Reference Consortium reference panel. Based on recommendations by the UKB we excluded all variants imputed from the UK10K reference panel, as technical errors may have occurred during the imputation process. The imputed data was converted to hard-called genotypes using a certainty threshold of 0.9.

**Neuroticism phenotype.** Neuroticism was measured using the Eysenck Personality Questionnaire, Revised Short Form (EPQ-R-S)[11], consisting of 12 dichotomous items ('yes' or 'no'). Participants completing <9 items were excluded from further analysis. We constructed a weighted neuroticism sum-score by adding up the individual item responses and dividing this sum by the total number of completed items (see Supplementary Tables 1, 2 and 5 for the endorsement rates of the individual items in both samples, as well as in the full sample). Prior to analysis, the sum-score was standardized to have mean 0 and variance 1.

**Cluster scores.** Scores for the depressed affect cluster were obtained by summing the scores on the four items 'Do you often feel lonely?', 'Do you ever feel 'just miserable' for no reason?', 'Does your mood often go up and down?', and 'Do you often feel 'fed-up'?'. Scores on the worry cluster were obtained by summing the scores on the four items 'Are you a worrier?', 'Do you suffer from nerves?', 'Would you call yourself a nervous person?', and 'Would you call yourself tense or highly strung'. Only participants with complete scores on all 4 items were included in GWA analyses of the cluster scores, resulting in sample sizes of $N = 361,768$ for depressed affect and $N = 350,356$ for worry.

**SNP-based analysis.** All genome-wide association (GWA) analyses were conducted separately on sample 1 and sample 2. GWA analyses of the 12 neuroticism items were conducted using logistic regression as implemented in PLINK 1.9[37,38], regressing the dichotomous neuroticism item responses on the imputed hard-called SNPs. Linear regression was used to analyze the neuroticism sum-score and both item cluster scores[19]. Sex, age, and townsend deprivation index (TDI; a measure

based on postal code indicating material deprivation), were included in the analyses as covariates. Genotype array was only included as covariate in the analyses of sample 1 as the same array was used for all subjects in sample 2. Additionally, we included the first 10 genetic PCs as covariates to control for potential population stratification. Genetic PCs were computed separately for both samples using FlashPCA 2[39] on individuals of European ancestry, after LD pruning and filtering out SNPs with MAF < 0.01 and genotype missingness > 0.05.

Final data analysis was restricted to autosomal, bi-allelic SNPs with MAF > 0.0001, high imputation quality (INFO score ≥0.9) and low missingness (<0.05), resulting in 10,847,151 SNPs. Within each SNP-based GWA analysis, we applied the standard genome-wide significance threshold in European-descent samples[40] of $P < 5 \times 10^{-8}$. In addition, we indicate whether the genetic signal survived correction for the 13 phenotypes tested ($P < 3.85 \times 10^{-9}$; Supplementary Data 2; Supplementary Fig. 4).

**Meta-analysis**. For all 13 phenotypes (12 items, sum), the SNP-based genome-wide association analyses were conducted separately on sample 1 and sample 2. To optimize the robustness of the genetic comparisons between the 13 phenotypes, we combined for all 13 phenotypes the GWA results from sample 1 and 2 using meta-analysis in METAL[41]. All SNP, gene, and gene-set analyses (Supplementary Data 2, 5–17, 18–30, 33–47) are thus based on the results from these meta-analyses.

**Excluded loci**. Based on visual inspection of the regional plots (LocusZoom) of all genomic loci identified in SNP-based meta-analyses of the 13 neuroticism phenotypes, we decided to further examine the plausibility of 6 of these loci, as they appeared to be driven by only one or two SNPs (Supplementary Data 3, 4). First, we compared the allele frequencies of the lead SNPs in these loci that we observed in the UKB data to the allele frequencies reported by 1000 genomes and TOPMED, as a large difference might indicate genotyping errors (Supplementary Data 3). Second, we compared the $P$ values from GWA analyses in both samples to see whether the signal was driven by a subset of the data. Finally, differential LD across datasets might be another indicator of genotyping errors. Therefore, we selected all SNPs in LD ($r^2 > 0.6$) with the lead SNPs in the 1000 Genomes data, and compared the LD in the 1000 Genomes data with LD (for the same combinations of SNPs) in the UKB data (Supplementary Data 4). None of these checks suggested genotyping errors of these SNPs. However, as the evidence for these loci was based on very few SNPs we decided to exclude them, and count the total number of unique genomic loci for all 12 items and the sum-score as 261–6 = 255.

**Gene-based analysis**. Gene-based genome-wide association analyses (GWGAS) were conducted in MAGMA (http://ctg.cncr.nl/software/magma23), using the $P$ values from the GWA analyses as input. This gene-based analysis tests the joint signal of all SNPs in a gene with the phenotype, while accounting for LD between those SNPs, thus uncovering gene-level signal that may go unnoticed in SNP-based analyses. In total, 18,183 genes were covered by at least one SNP, and used in the GWGAS. Within each gene-based analysis, we applied a stringent Bonferroni correction for the number of tested genes, resulting in a genome-wide threshold for significance of $P < 2.75 \times 10^{-6}$ (0.05/number of genes tested). In addition, we indicate whether the gene-signal survives correction for the 13 phenotypes tested ($P < 2.12 \times 10^{-7}$; Supplementary Data 33; Supplementary Fig. 22).

**Gene-set analysis**. Using MAGMA[23], we tested the association between all 13 neuroticism phenotypes and 7,297 gene-sets. Specifically, we tested for association with 7,244 gene ontology gene-sets (MsigDB version 6.0; http://software.broadinstitute.org/gsea/msigdb/collections.jsp) and 53 tissue expression profiles (https://www.gtexportal.org/home/). For all gene-sets competitive $P$ values were computed, which result from the test whether the combined effect of genes in a gene-set is significantly larger than the combined effect of a same number of randomly selected genes (in contrast, self-contained $P$ values result from testing against the null hypothesis of no effect). We only report competitive $P$ values, which are more conservative compared to self-contained $P$ values. Competitive $P$ values were Bonferroni corrected (α = 0.05/7,297 = 6.85 × 10$^{-6}$).

**SNP-based heritability and genetic correlations**. LD score regression (LDSC[16,17]) analyses were run on summary statistics obtained from GWA analyses to estimate the proportion of phenotypic variance attributable to all SNPs in the analyses (SNP-based heritability; $h^2_{SNP}$). Bivariate LD Score regression (https://github.com/bulik/ldsc) on the GWAS summary statistics was used to estimate the genetic correlations between all neuroticism phenotypes. Significance of genetic correlations between all neuroticism phenotypes was determined by applying a Bonferroni corrected threshold of $P = 2.22 \times 10^{-4}$ (0.05/(15 × 15) = 0.05/225; Fig. 1 and Supplementary Figure 24 combined).

The genetic correlations between the 15 neuroticism phenotypes and 33 sundry traits for which summary statistics were available, were also calculated using LD score regression. We corrected for multiple testing through a stringent Bonferroni correction leading to a $P$ value threshold of $P < 1.01 \times 10^{-4}$ (0.05/(15 × 33) = 0.05/495).

**Genetic overlap tests for top associated SNPs**. Fisher's exact tests were used to examine the overlap in the top associated SNPs across all pairwise combinations of neuroticism items and the sum-score. Prior to this analysis, we applied LD-based pruning on all items to ensure that SNPs were independent (a subset of 10,000 UKB participants was used as a reference set; $r^2 = 0.8$). Sign tests were used to establish whether signs of top associated SNPs were in the same direction across all neuroticism items and the sum-score. Here we used clumping ($r^2 = 0.1$ with window size 500 kb) based on the association $P$ values to ensure that the most strongly associated variants were not lost. As clumping results in different SNP sets for different items, sign test results are not symmetrical for pairs of items. For both tests, we used multiple $P$ value thresholds, including the conventional GWS threshold ($5 \times 10^{-8}$) and a conservative threshold correcting for the 13 phenotypes tested ($3.85 \times 10^{-9}$).

We acknowledge that smoother measures than thresholding are available. For example, the Rank-Rank-Hypergeometric-Overlap (RRHO) test[42] displays overlap in a more nuanced manner. However, as we generally worked with 13 phenotypes, resulting in 13 × 13–13 = 156 combinations (considering that some of the thresholded tests were not symmetrical), a method like RRHO would yield 156 individual figures from which it would be difficult to discern patterns in the results.

**Functional annotation and gene-mapping using FUMA**. The FUMA GWAS platform (http://fuma.ctglab.nl/20) uses GWAS summary statistics to functionally map, annotate, prioritize, visualize, and interpret GWAS results. We used the summary statistics from GWA meta-analyses on the 12 individual items and the neuroticism sum-score as input for FUMA.

FUMA first defined independent significant SNPs which have a genome-wide significant $P$ value ($5 \times 10^{-8}$) and are independent at $r^2 < 0.6$. Subsequently, lead SNPs were defined by retaining those independent significant SNPs that were independent from each other at $r^2 < 0.1$. Next, risk loci were defined by merging physically overlapping lead SNPs or lead SNPs whose LD blocks were closer than 250 kb apart. A consequence of this definition of risk loci is that the same locus may be discovered for different phenotypes included in the study, while the lead SNPs are different.

All SNPs in LD ( > 0.6) with one of the independent significant SNPs were used in annotation. Functional consequences were obtained by performing ANNOVAR gene-based annotation using Ensembl genes. In addition, potential regulatory functions are indicated by the RegulomeDB score[43] (with lower scores indicating a higher probability of having a regulatory function) and by 15-core chromatin states predicted by ChromHMM[44] for 127 tissue/cell types[45].

All SNPs in genomic risk loci that were GWS, or in LD ( > 0.6) with one of the independent GWS SNPs were mapped to genes in FUMA[20] using either of three strategies.

The first strategy we applied, positional mapping, was used to map SNPs to genes based on the physical distances (i.e., within 10 kb window) from known protein coding genes in the human reference assembly (GRCh37/hg19).

The second strategy, eQTL mapping, is used to link SNPs to genes with which these SNPs show a significant eQTL association (i.e., allelic variation at the SNP affects the expression of that gene). This strategy is based on information from 3 data repositories (GTEx, Blood eQTL browser, and BIOS QTL browser), and uses cis-eQTLs, which can map SNPs to genes that lie up to 1 Mb apart. We applied a false discovery rate (FDR) of 0.05 to define significant eQTL associations.

Finally, using chromatin interaction, SNPs were mapped to genes based on a significant chromatin interaction between a genomic region in a risk locus and promoter regions of genes (250 bp up- and 500 bp downstream of transcription start site (TSS)). Unlike eQTL mapping, chromatin interaction mapping has no distance boundary and can involve long-range interactions. Currently, Hi-C data of 14 tissue types are included in FUMA[46]. Generally, chromatin interactions are defined in a certain resolution (40 kb in this case) such that interacting regions may span multiple genes. All SNPs within these regions would be mapped by this method to genes in the corresponding interaction region. To further prioritize candidate genes from chromatin interaction mapping, we integrated predicted enhancers and promoters in 111 tissue/cell types from the Roadmap Epigenomics Project[45]; chromatin interactions are selected in which one region involved in the interaction overlaps with predicted enhancers and the other region overlaps with predicted promoters in 250 bp upstream and 500 bp downstream of the TSS site of a gene. A FDR of $1 \times 10^{-5}$ was applied to define significant interactions.

**Data availability**. Our policy is to make genome-wide summary statistics (sumstats) publicly available. Sumstats from the GWA analyses of all neuroticism phenotypes are available for download at https://ctg.cncr.nl/.

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

## Acknowledgements

This work was funded by The Netherlands Organization for Scientific Research (NWO MagW VIDI 452-12-014; NWO VICI 453-14-005). The analyses were carried out on the Genetic Cluster Computer, which is financed by the Netherlands Organization for Scientific Research (NWO: 480-05-003), by the VU University (Amsterdam, The Netherlands), and by the Dutch Brain Foundation, and is hosted by the Dutch National Computing and Networking Services SurfSARA. This research has been conducted using the UK Biobank Resource, application number 16406. We thank the participants and researchers who collected and contributed to the data. Summary statistics will be made available for download from http://ctglab.vu.nl

## Author contributions

M.N. performed the analyses. S.vd.S, D.P., and M.N. conceived the study. S.S. QC-ed the UKB data and developed the analysis pipeline. K.W. constructed the FUMA tool for biological annotation. M.N., S.vd.S, and D.P. wrote the paper. All authors discussed the results and commented on the paper.

## Additional information

**Competing interests:** The authors declare no competing financial interests.

