## [Peer Review File · Nature Communications]

Reviewer #1 (Remarks to the Author):

I must admit I read this report with great interest and pleasure. It does investigate a very important subject - the genetic architecture of neuroticism and its components - and as a conceptual approach it might be even more important. It was performed on the UK Biobank (prior release) which is a large sample, and as such samples go, it is very homogenous in its measures.

The study has been performed very diligently and is very well described - and indeed the claims made are well supported by the data.

I only have two points where I would think more work would be very helpful and could strengthen the paper -

a) replication - as far as I can see this analysis is done only in UK Biobank - thus we simply are not protected from the results being specific to this sample only. A replication - even partially on some of the measures would greatly relieve this reviewer.

b) thresholding - at times the authors do threshold SNPs to assign properties - e.g. in the input to supp fig 11. I am aware that this is a very common procedure but it is a very crude way to go - I suppose that smoother measures on joint information might do more justice to the data and provide more power.

Reviewer #2 (Remarks to the Author):

Using data from the UK biobank (ns 107,462-110,197), this study conducted a GWAS of individual neuroticism items (n=12) from the EPQ-R as well as its sum score. In addition to identifying 22 independent SNPs associated with these neuroticism metrics (some items showed no association), and 54 genes through gene-based analyses, great heterogeneity in the individual SNPs associated with specific items was observed, which was also reflective in broad heterogeneous genetic correlations derived from LD score regression (0.31 – 0.92). Clustering analyses identified 2 genetically homogenous subsets (i.e., 1: worry/suf-nerv, nerve-feel, tense and 2: lone, mis, mood, fed-up) that correspond to the worry and depressed affect scales through factor analysis of the EPQ-R. I was very favorably impressed by this study: it is well powered, well-written and importantly contributes to a timely debate regarding phenotypic precision in psychiatric genetics by showing item-specific correlations and heterogeneity in the genetic architecture of a construct. Below are suggestions that I believe would further improve this already excellent manuscript.

1. Traditional GWAS significance threshold, $5E-8$, was used to evaluate significance levels. However, 13 GWAS (of each item + sum scores) were conducted. It would thus appear that additional multiple

testing correction should be implemented (arguably this could be less severe than a strict bonferroni given the phenotypic correlation). It would be helpful if the authors could note what signals would survive correction for the 13 GWASs run (e.g., worry and signal from 8q23.1)

2. A post-hoc GWAS of the 2 identified homogenous subsets may be useful to report in the supplement. Are there more genetic signals in these relative to the single items or the sum total? Is there more consistency and divergence across functional gene sets? I.e., could such data be used to guide phenotypic characterization in future GWAS? Does the depressed affect subset correlate more with MDD and the worry aspect correlate more with anxiety disorders?

3. Where GTex analyses restricted to those of European ancestry? The authors may find it beneficial to also replicate such expression analyses using BRAINEAC database that is freely available online (<http://www.braineac.org/>).

4. Some SNP level annotation for the independent SNPs would be helpful. For example, regulomeDB provides a score of 1f for rs12547493 suggesting it may alter binding, and expression. Adding such material would further improve the impact of this manuscript.

Amsterdam, 5 December 2017

Responses to reviewers

General comment: The initial version of this manuscript did not follow Nature Communications' formatting guidelines. As per the editor's suggestion, we have made use of the more lenient word allowance of *Nature Communications*, which allowed us to explain analyses and results more thoroughly, and to extend the introduction and discussion. Together with Reviewer 1's request to add a replication sample, this resulted in large portions of text that are new, that have been added for clarification, or that have been moved from the supplementary to the main.

Below we reproduce the reviewers' points below (italics), together with the details of how we have addressed them (normal blue font).

Reviewer #1 (Remarks to the Author):

I must admit I read this report with great interest and pleasure. It does investigate a very important subject - the genetic architecture of neuroticism and its components - and as a conceptual approach it might be even more important. It was performed on the UK Biobank (prior release) which is a large sample, and as such samples go, it is very homogeneous in its measures.

The study has been performed very diligently and is very well described - and indeed the claims made are well supported by the data.

I only have two points where I would think more work would be very helpful and could strengthen the paper.

a) replication - as far as I can see this analysis is done only in UK Biobank - thus we simply are not protected from the results being specific to this sample only. A replication - even partially on some of the measures would greatly relieve this reviewer.

We agree with the reviewer that replication greatly enhances the reliability and credibility of the results. In this study, convincing replication requires an independent, sufficiently large, dataset containing the same item-level data. Unfortunately, very few item- and symptom-level datasets are available. For the current revision, we therefore decided to make use of the UK Biobank data that became available in the newest release (July 2017), as the complementary data set is of sufficient size and concerns the exact same phenotypes.

Using two samples from the UK biobank (i.e., sample 1 (as used in the original submission): first UK Biobank release, N= 106,218–109,017; sample 2 (participants added in the second UK Biobank release): N = 260,083–266,896 independent from sample 1), we were now able to replicate the GWA analyses on the individual Neuroticism items, and to compute genetic correlations between the items within and across samples, thereby replicating a) the observed heterogeneity in the items and b) the pattern of inter-items genetic correlations which indicated the existence of two relatively homogeneous item clusters.

We show that the directions of SNP effects were almost identical across the two samples (i.e., 94% of the SNPs with $P < 0.05$ has the same direction of effect, for all 15 phenotypes). We subsequently meta-analyzed the results obtained from sample 1 and sample 2 for each of the 13 phenotypes (12 items, sum score). For all subsequent analyses (e.g., gene-based tests, gene-set tests, functional annotation, genetic correlations with external variables) we used the meta-analysis results as input for two reasons: 1) discussing all these analyses for the two samples separately quickly becomes

intractable, and 2) the power to robustly detect genetic effects for all phenotypes, especially when Bonferroni-correcting for the number of tested phenotypes (see Reviewer 2, point 1), is much better in the combined sample.

The choice to replicate within the UK Biobank sample considerably improved the robustness and credibility of the item-level analyses.

We note explicitly that we redid all original analyses for sample 1 as well because with the new UK Biobank release, newly imputed genetic data were available for the full UK Biobank sample on approximately 96 million genetic variants.

To accommodate these changes, we made the following changes:

- added text on p.5 – p.7 (line 75 – 104): [“To replicate these findings” ... “the sum-score^{3,4,21,22} (Supplementary Fig. 18a).”]
- added text in the Methods on p.14 (line 267 – 273): [“503,325 participants” ... “Supplementary Table 2).”]
- added text in the Methods on p.14 – p.15 (line 278 – 295): [“Therefore, principal components” ... “using a certainty threshold of 0.9.”]
- added text in the Methods on p.15 – p.16 (line 302 – 318): [“(see Supplementary Tables 1-2 & 6” ... “sample 1 and sample 2.”]
- added text in the Methods on p.16 – p.17 (line 335 – 357): [“Meta-analysis” ... “all 12 items and the sum-score as 261-6=255.”]
- changed Fig. 1
- changed Supplementary Tables 1-22

b) thresholding - at times the authors do threshold SNPs to assign properties - e.g. in the input to suppl. fig 11. I am aware that this is a very common procedure but it is a very crude way to go - I suppose that smoother measures on joint information might do more justice to the data and provide more power.

This comment specifically concerned the two supplementary figures on the sign-concordance test and the Fisher exact test. We agree with the reviewer that smoother procedures than simple thresholding could yield less crude information. For example, the Rank-Rank-Hypergeometric-Overlap (RRHO) test (Franke et al, Nature Neuroscience, 2016) could be used to display the overlap in a more nuanced manner. However, as we generally worked with 13 phenotypes, resulting in $13 \times 13 = 169$ combinations (considering that some of the thresholded tests were not symmetrical), a method like RRHO would yield 169 individual figures from which it would be difficult to discern patterns in the results. In contrast, using the thresholded tests we now obtained 4 figures (for 4 thresholds), which do allow us to discern patterns in the results. For reasons of parsimony and efficiency, we, therefore, chose to adhere to the conventional method of thresholding, while acknowledging its drawbacks (p.19 – p.20, line 408 – 413). We did, however, add a more stringent threshold, i.e., Bonferroni-corrected for the 13 phenotypes, as requested by Reviewer 2 (point 1).

We revised Supplementary Figs. 20 and 21 (previously Supplementary Figs. 8 and 9) accordingly.

To accommodate these changes, we adjusted the following section on p.7 (line 110 – 115): [“In studying overlap” ... “considerably between individual measures.”]

Reviewer #2 (Remarks to the Author):

Using data from the UK biobank (ns 107,462-110,197), this study conducted a GWAS of individual neuroticism items (n=12) from the EPQ-R as well as its sum score. In addition to identifying 22 independent SNPs associated with these neuroticism metrics (some items showed no association), and 54 genes through gene-based analyses, great heterogeneity in the individual SNPs associated with specific items was observed, which was also reflective in broad heterogeneous genetic correlations derived from LD score regression (0.31 – 0.92). Clustering analyses identified 2 genetically homogenous subsets (i.e., 1: worry/suf-nerv, nerve-feel, tense and 2: lone, mis, mood, fed-up) that correspond to the worry and depressed affect scales through factor analysis of the EPQ-R. I was very favorably impressed by this study: it is well powered, well-written and importantly contributes to a timely debate regarding phenotypic precision in psychiatric genetics by showing item-specific correlations and heterogeneity in the genetic architecture of a construct. Below are suggestions that I believe would further improve this already excellent manuscript.

1. Traditional GWAS significance threshold, $5E-8$, was used to evaluate significance levels. However, 13 GWAS (of each item + sum score) were conducted. It would thus appear that additional multiple testing correction should be implemented (arguably this could be less severe than a strict bonferroni given the phenotypic correlation). It would be helpful if the authors could note what signals would survive correction for the 13 GWASs run (e.g., worry and signal from 8q23.1).

We agree with the reviewer that it is important to indicate which signals remain after applying correction for multiple testing, i.e., testing 13 phenotypes (12 items and the sum-score). Although correcting for the total number of phenotypes would indeed likely be too strict (because of intercorrelations), we rather err on the side of caution, and now included a strict Bonferroni corrected P value threshold: 3.85×10^{-9} ($5 \times 10^{-8}/13$) for SNP-based testing, and 2.12×10^{-7} ($(0.05/\text{no. of genes tested})/13$) for gene-based testing. As proposed by the reviewer, we have, for SNP- as well as for gene-based results, indicated in the respective tables whether the genetic signal would survive this conservative multiple testing correction (see Suppl. Tables 7 & 38, Suppl. Figs. 4 & 22). When comparing results between phenotypes in the main text of the manuscript, we used, as in the original submission, the standard threshold for genome-wide significance (5×10^{-8} , as explicitly noted in the Methods on pages 16-18). We do so to ensure comparability with other (published) GWA studies on e.g. the Neuroticism sum-score.

We note that as the sample size in the full sample (sample 1 + sample 2) used for the current revision is substantially higher than the sample size in the original version of the manuscript, considerable signal remains after applying the Bonferroni correction. This increased our confidence in the results, as item- and cluster-specific signal was still clearly observed. We also note that the increase in sample size increased the number of significant findings to such an extent that the genome-wide significant SNPs and genes could no longer be included in tables in the main text: this information is now provided in Supplementary Tables 7 and 38.

In addition, we added the following on p.16 (line 331 – 333):

["In addition, we indicate whether the genetic signal survived correction for the 13 phenotypes tested ($P < 3.85 \times 10^{-9}$; Supplementary Table 7; Supplementary Fig. 4)."]

And on p.18 (line 367 – 369):

["In addition, we indicate whether the gene-signal survives correction for the 13 phenotypes tested ($P < 2.12 \times 10^{-7}$; Supplementary Table 38; Supplementary Fig. 22)."]

2. A post-hoc GWAS of the 2 identified homogenous subsets may be useful to report in the supplement. Are there more genetic signals in these relative to the single items or the sum total? Is there more consistency and divergence across functional gene sets? I.e., could such data be used to guide phenotypic characterization in future GWAS? Does the depressed affect subset correlate more with MDD and the worry aspect correlate more with anxiety disorders?

We appreciate this suggestion. The current paper aims to show that item-level analyses can yield different outcomes when compared to sum-score analysis, which can be ascribed to item-level heterogeneity. We show that subsets of items can be used to create genetically homogenous clusters. These results indeed suggest that such homogenous clusters may be used in future GWAS aimed at neuroticism or its overlap with MDD. After our initial submission of the current paper on the UKB1 sample, we continued with these outcomes using the full UKB sample as soon as it became available and used the two homogenous sub-clusters in a second paper, which specifically addresses Neuroticism composite scores, available on bioRxiv since Sept 5 (Nagel et al., 2017, <https://www.biorxiv.org/content/early/2017/09/05/184820>). In Nagel et al. (2017), we report extensively on functional consequences of the two clusters in comparison with the sum score and also compared with MDD. This second paper does not include the justification of using these two clusters, nor does it include item-level results. The second paper focusses on gene finding for composite scores, whereas the current paper focuses on item heterogeneity and item specific findings, providing the empirical justification of the two clusters. We feel that we cannot include extensive functional annotation of, or biological comparison between, the two clusters in the current paper, as this is already part of the second paper that followed from the conclusions of the current paper.

We note, however, that in the newly included Table 1, we do highlight a selection of interesting genes, some of which were clearly cluster-specific.

3. Were GTex analyses restricted to those of European ancestry? The authors may find it beneficial to also replicate such expression analyses using BRAINEAC database that is freely available online (<http://www.braineac.org/>).

The reviewer suggested to replicate the GTEx results in BRAINEAC from the UK Brain Expression Consortium (UKBEC). In following this suggestion, we first calculated the correlation of gene expression in specific tissues between GTEx (RNA sequencing) and the UKBEC data (exon array). This correlation was approximately zero, indicating that any patterns observed in the GTEx data were unlikely to also occur in the UKBEC data, and vice versa. We suspect large batch effects between both datasets, which we cannot control for as we only have access to the normalized expression values for the UKBEC data. The alarmingly low correlation between the two data sets made us doubt the robustness and generalizability of our results to such an extent that we eventually decided to altogether exclude this co-expression analysis from the paper.

4. Some SNP level annotation for the independent SNPs would be helpful. For example, regulomeDB provides a score of 1f for rs12547493 suggesting it may alter binding, and expression. Adding such material would further improve the impact of this manuscript.

We thank the reviewer for this suggestion. We have now added a section on functional follow-up analyses, which includes a discussion on exonic non-synonymous (ExNS) SNPs identified for specific items or the sum-score. The supplementary materials now also feature a table with all ExNS SNPs identified for individual items or the sum-score, in which we also report on regulome DB scores and minimum chromatin state for these SNPs (Supplementary Table 56).

We also added a table (Table 1, main text) highlighting a selection of “global” genes (affecting most or all of the 13 phenotypes) and “local” genes (only affecting 1 of the items specifically).

We added the following on p.9, discussing exonic non-synonymous (ExNS) SNPs identified for specific items or the sum-score (line 152 – 170):

[“Functional follow-up” ... “affect binding and to affect expression of a gene target.”]

In the main text, we describe this on p.8 (line 128 – 132):

[“Overall, the results from the gene-based analyses” ... “were *not* identified in analysis of the sum-score (Supplementary Figs. 18d).”]

Reviewer #1 (Remarks to the Author):

I have no further objections, all my comments have been answered satisfactorily

Reviewer #2 (Remarks to the Author):

The authors were incredibly responsive to reviewer comments and I no longer have any substantive concerns about the paper.

In particular the addition of a replication sample significantly enhanced this paper.